# Unbalanced Sobolev Descent

**Youssef Mroueh, Mattia Rigotti**
IBM Research AI
mroueh@us.ibm.com, mrg@zurich.ibm.com

## Abstract

We introduce Unbalanced Sobolev Descent (USD), a particle descent algorithm for transporting a high dimensional source distribution to a target distribution that does not necessarily have the same mass. We define the *Sobolev-Fisher* discrepancy between distributions and show that it relates to advection-reaction transport equations and the Wasserstein-Fisher-Rao metric between distributions. USD transports particles along gradient flows of the witness function of the Sobolev-Fisher discrepancy (advection step) and reweighs the mass of particles with respect to this witness function (reaction step). The reaction step can be thought of as a birth-death process of the particles with rate of growth proportional to the witness function. When the *Sobolev-Fisher witness* function is estimated in a Reproducing Kernel Hilbert Space (RKHS), under mild assumptions we show that USD converges asymptotically (in the limit of infinite particles) to the target distribution in the Maximum Mean Discrepancy (MMD) sense. We then give two methods to estimate the *Sobolev-Fisher witness* with neural networks, resulting in two Neural USD algorithms. The first one implements the reaction step with mirror descent on the weights, while the second implements it through a birth-death process of particles. We show on synthetic examples that USD transports distributions with or without conservation of mass faster than previous particle descent algorithms, and finally demonstrate its use for molecular biology analyses where our method is naturally suited to match developmental stages of populations of differentiating cells based on their single-cell RNA sequencing profile. Code is available at http://github.com/ibm/usd.

## 1 Introduction

Particle flows such as Stein Variational Gradient descent [1], Sobolev descent [2] and MMD flows [3], allow the transport of a source distribution to a target distribution, following paths that progressively decrease a discrepancy between distributions (Kernel Stein discrepancy and MMD, respectively). Particle flows can be seen through the lens of Optimal Transport as gradient flows in the Wasserstein geometry [4], and they've been recently used to analyze the dynamics of gradient descent in over-parametrized neural networks in [5] and of Generative Adversarial Networks (GANs) training [2].

Unbalanced Optimal Tansport [6, 7, 8, 9] is a new twist on the classical Optimal Transport theory [10], where the total mass between source and target distributions may not be conserved. The Wasserstein Fisher-Rao (WFR) distance introduced in [7] gives a dynamic formulation similar to the so-called Benamou-Brenier dynamic form of the Wasserstein-2 distance [11], where the dynamics of the transport is governed by an advection term with a velocity field $V_t$ and a reaction term with a rate of growth $r_t$, corresponding to the construction and destruction of mass with the same rate:

$$\text{WFR}^2(p,q) = \inf_{q_t, V_t, r_t} \int_0^1 \int (\|V_t(x)\|^2 + \frac{\alpha}{2} r_t^2(x)) dq_t(x) dt$$

$$\text{s.t } \frac{\partial q_t(x)}{\partial t} = -\text{div}(q_t(x)V_t(x)) + \alpha \, r_t(x) q_t(x), \qquad q_0 = q, q_1 = p. \tag{1}$$

From a particle flow point of view, this advection-reaction in Unbalanced Optimal Transport corresponds to processes of birth and death, where particles are created or killed in the transport from source to target. Particle gradient descent using the WFR geometry have been used in the analysis of over-parameterized neural networks and implemented as Birth-Death processes in [12] and as conic descent in [13]. In the context of particles transportations, [14] showed that birth and death processes can accelerate the Langevin diffusion. On the application side, Unbalanced Optimal Transport is a powerful tool in biological modeling. For instance, the trajectories of a tumor growth have been modeled in the WFR framework by [15]. [16] and [17] used Unbalanced Optimal Transport to find differentiation trajectories of cells during development.

The dynamic formulation of WFR is challenging as it requires solving PDEs. One can use the unbalanced Sinkhorn divergence and apply an Euler scheme to find the trajectories between source and target as done in [18] but this does not give any convergence guarantees.

In this paper we take another approach similar to the one of Sobolev Descent [2]. We introduce the Kernel Sobolev-Fisher discrepancy that is related to WFR and has the advantage of having a closed form solution. We present a particle descent algorithm in the unbalanced case named *Unbalanced Sobolev Descent* (USD) that consists of two steps: an advection step that uses the gradient flows of a witness function of the Sobolev-Fisher discrepancy, and a reaction step that reweighs the particles according to the witness function. We show theoretically that USD is convergent in the Maximum Mean Discrepancy sense (MMD), that the reaction step accelerates the convergence, in the sense that it results in strictly steeper descent directions, and give a variant where the witness function is efficiently estimated as a neural network. We then empirically demonstrate the effectiveness and acceleration of USD in synthetic experiments, image color transfer tasks, and finally use it to model the developmental trajectories of populations of cells from single-cell RNA sequencing data [16].

## 2 Sobolev-Fisher Discrepancy

In this Section we define the Sobolev-Fisher Discrepancy (SF) and show how it relates to advection-reaction PDEs. While this formulation remains computationally challenging, we'll show in Section 3 how to approximate it in RKHS.

### 2.1 Advection-Reaction with no Conservation of Mass

**Definition 1** (Sobolev-Fisher Discrepancy). *Let $p, q$ be two measures defined on $\mathcal{X} \subset \mathbb{R}^d$. For $\alpha > 0$, the Sobolev-Fisher Discrepancy is defined as follows:*

$$SF(p,q) = \sup_f \left\{ \mathbb{E}_{x \sim p} f(x) - \mathbb{E}_{x \sim q} f(x) : \quad \mathbb{E}_{x \sim q} \|\nabla_x f(x)\|^2 + \alpha \mathbb{E}_{x \sim q} f^2(x) \le 1, \quad f|_{\partial \mathcal{X}} = 0 \right\}$$

Note that the objective of SF is an Integral Probability Metric (IPM) objective, and the function space imposes a constraint on the weighted Sobolev norm of the witness function $f$ on the support of the distribution $q$. We refer to $q$ as the source distribution and $p$ as the target distribution. The following theorem relates the solution of the Sobolev-Fisher Discrepancy to an advection-reaction PDE:

**Theorem 1** (Sobolev-Fisher Critic as Solution of an Advection-Reaction PDE). *Let $u$ be the solution of the* advection-reaction *PDE:*

$$p(x) - q(x) = -div(q(x)\nabla_x u(x)) + \alpha u(x)q(x), \quad u|_{\partial \mathcal{X}} = 0.$$

*Then $SF^2(p,q) = \mathbb{E}_{x \sim q} \|\nabla_x u(x)\|^2 + \alpha \mathbb{E}_{x \sim q} u^2(x)$, with witness function $f_{p,q}^* = u/SF(p,q)$.*

From Theorem 1 we see that the witness function of $SF^2$ solves an advection-reaction where the mass is transported from $q$ to $p$, via an advection term following the gradient flow of $\nabla_x u$, and a reaction term amounting to construction/destruction of mass that we also refer to as a birth-death process with a rate given by $u$. Intuitively, if the witness function $u(x) > 0$ we need to create mass, and destruct mass if $u(x) < 0$. This is similar to the notion of particle birth and death defined in [12] and [14].

In Proposition 1 we give a convenient unconstrained equivalent form for $SF^2$:

**Proposition 1** (Unconstrained Form of $SF^2$). *SF satisfies the expression: $SF^2(p,q) = \sup_u L(u)$, with $L(u) = 2(\mathbb{E}_{x \sim p} u(x) - \mathbb{E}_{x \sim q} u(x)) - \left( \mathbb{E}_{x \sim q} \|\nabla_x u(x)\|^2 + \alpha \mathbb{E}_{x \sim q} u^2(x) \right).$*

Theorem 2 gives a physical interpretation for $SF^2$ as finding the witness function $u$ that has minimum sum of kinetic energy and rate of birth-death while transporting $q$ to $p$ via advection-reaction:

**Theorem 2** (Kinetic Energy & Birth-Death rates minimization). *Consider the following minimization:*

$$P = \inf_{\substack{V:\mathcal{X}\to\mathbb{R}^d \\ r:\mathcal{X}\to\mathbb{R}}} \left\{ \frac{1}{2} \left( \int_{\mathcal{X}} (\|V(x)\|^2 + \alpha r^2(x))q(x)dx \right) : p(x) - q(x) = -div(q(x)V(x)) + \alpha r(x)q(x) \right\}$$

*We then have that $P = \frac{1}{2}SF^2(p,q)$, and moreover:*

$$SF^2(p,q) = \inf_u \int_{\mathcal{X}} \|\nabla_x u(x)\|^2 q(x)dx + \alpha \int_{\mathcal{X}} u^2(x)q(x)dx,$$

$$\text{subject to } p(x) - q(x) = -div(q(x)\nabla_x u(x)) + \alpha u(x)q(x).$$

**Remarks.** a) When $\alpha = 0$ we obtain the Sobolev Discrepancy, or $\|p - q\|_{\dot{H}^{-1}(q)}$, that linearizes the Wasserstein-2 distance. b) Note that this corresponds to a Beckman type of optimal transport [19], where we transport $q$ to $p$ ($q$ and $p$ do not have the same total mass) via an advection-reaction with mass not conserved. It is easy to see that $\int_{\mathcal{X}}(p(x) - q(x))dx = \alpha \int_{\mathcal{X}} u(x)q(x)dx$.

## 2.2 Advection-Reaction with Conservation of Mass

Define the Sobolev-Fisher Discrepancy with conservation of mass: $\overline{SF}^2(p,q) = \sup_u L(u)$, where

$$L(u) = 2\left(\mathbb{E}_{x\sim p}u(x) - \mathbb{E}_{x\sim q}u(x)\right) - \left(\mathbb{E}_{x\sim q}\|\nabla_x u(x)\|^2 + \alpha\left(\mathbb{E}_{x\sim q}\left(u(x) - \mathbb{E}_{x\sim q}(u(x))\right)^2\right)\right).$$

The only difference between the previous expression and $SF^2$ in Proposition 1 is that the variance of the witness function is kept under control, instead of the second order moment. Defining

$$\mathcal{E}(u) = \int_{\mathcal{X}} (\|\nabla_x u(x)\|^2 + \alpha(u(x) - \mathbb{E}_{x\sim q}u(x))^2)q(x)dx$$

one can similarly show that $\overline{SF}$ has the primal representation:

$$\overline{SF}^2(p,q) = \inf_u \left\{\mathcal{E}(u) : p(x) - q(x) = -div(q(x)\nabla_x u(x)) + \alpha(u(x) - \mathbb{E}_{x\sim q}u(x))q(x)\right\}.$$

Hence, we see that $\overline{SF}$ is the minimum sum of kinetic energy and variance of birth-death rate for transporting $q$ to $p$ following an advection-reaction PDE with conserved total mass. The conservation of mass comes from the fact that $\chi(x) = -div(q(x)\nabla_x u(x)) + \alpha(u(x) - \mathbb{E}_{x\sim q}u(x))q(x)$ satisfies:

$$\int_{\mathcal{X}}(p(x) - q(x))dx = \int_{\mathcal{X}} \chi(x) = 0.$$

# 3 Kernel Sobolev-Fisher Discrepancy

In this section we turn to the estimation of SF discrepancy by restricting the witness function to a Reproducing Kernel Hilbert Space (RKHS), resulting in a closed-form solution.

## 3.1 Estimation in Finite Dimensional RKHS

Consider the finite dimensional RKHS, corresponding to an $m$ dimensional feature map $\Phi$:

$$\mathcal{H} = \{f \,|\, f(x) = \langle w, \Phi(x)\rangle \text{ where } \Phi : \mathcal{X} \to \mathbb{R}^m, w \in \mathbb{R}^m\}.$$

Define the kernel mean embeddings $\mu(p) = \mathbb{E}_{x\sim p}\Phi(x), \mu(q) = \mathbb{E}_{x\sim q}\Phi(x)$, and $\delta_{p,q} = \mu(p) - \mu(q)$. Let $C(q) = \mathbb{E}_{x\sim q}\Phi(x) \otimes \Phi(x)$ be the covariance matrix and $D(q) = \mathbb{E}_{x\sim q}J\Phi(x)^\top J\Phi(x)$ be the Gramian of the Jacobian, where $[J\Phi(x)]_{a,j} = \frac{\partial\Phi_j(x)}{\partial x_a}, a = 1\ldots d, j = 1\ldots m$.

**Definition 2** (Regularized Kernel Sobolev-Fisher Discrepancy (KSFD)). *Let $u \in \mathcal{H}$, and let $\lambda > 0$ and $\gamma \in \{0,1\}$, define: $L_{\gamma,\lambda}(u) = 2(\mathbb{E}_{x\sim p}u(x) - \mathbb{E}_{x\sim q}u(x)) - \left(\mathbb{E}_{x\sim q}[\|\nabla_x u(x)\|^2 + \alpha(u(x) - \gamma\mathbb{E}_q u(x))^2] + \lambda\|u\|_{\mathcal{H}}^2\right)$. The Regularized Kernel Sobolev-Fisher Discrepancy is defined as:*

$$SF^2_{\mathcal{H},\gamma,\lambda}(p,q) = \sup_{u\in\mathcal{H}} L_{\gamma,\lambda}(u).$$

*When $\gamma = 0$ this corresponds to the unbalanced case, i.e. birth-death with no conservation of total mass, while for $\gamma = 1$ we have birth-death with conservation of total mass.*

**Proposition 2** (Estimation in RKHS). *The Kernel Sobolev-Fisher Discrepancy is given by:* $SF^2_{\mathcal{H},\gamma,\lambda}(p,q) = \langle u^{\lambda,\gamma}_{p,q}, \delta_{p,q} \rangle$, *where the critic* $u^{\lambda,\gamma}_{p,q} = (D(q) + \alpha C_\gamma(q) + \lambda I_m)^{-1} \delta_{p,q}$, *with* $C_\gamma(q) = C(q) - \gamma \mu(q)\mu(q)^\top$. *Let* $u^{\lambda,\gamma}_{p,q}(x) = \langle u^{\lambda,\gamma}_{p,q}, \Phi(x) \rangle$ *and* $\delta_{p,q}(x) = \langle \delta_{p,q}, \Phi(x) \rangle$, *then:* $\nabla_x u^{\lambda,\gamma}_{p,q}(x) = (D(q) + \alpha C_\gamma(q) + \lambda I_m)^{-1} \nabla_x \delta_{p,q}(x)$.

**Remarks.**   a) For the unbalanced case $\gamma = 0$, we refer to $SF^2_{\mathcal{H},0,\lambda}$ as $SF^2_{\mathcal{H},\lambda}$. For the case of mass conservation $\gamma = 1$, refer to $SF^2_{\mathcal{H},1,\lambda}$ as $\overline{SF}^2_{\mathcal{H},\lambda}$. Note that $C_1(q) = \bar{C}(q) = C(q) - \mu(q)\mu(q)^\top$. b) A similar Kernelized discrepancy was introduced in [20], but not as an approximation of the Sobolev-Fisher discrepancy, nor in the context of unbalanced distributions and advection-reaction. c) For $\alpha = 0$ we obtain the kernelized Sobolev Discrepancy KSD of [2].

## 3.2   Kernel SF for Direct Measures

Consider direct measures $p = \sum_{i=1}^N a_i \delta_{x_i}$ and $q = \sum_{j=1}^n b_j \delta_{y_j}$ (with no conservation of mass we can have $\sum_i a_i \neq \sum_j b_j \neq 1$ ). An estimate of the Sobolev-Fisher critic is given by $\hat{u}^{\lambda,\gamma}_{p,q} = (\hat{D}(q) + \alpha \hat{C}_\gamma(q) + \lambda I_m)^{-1}(\hat{\mu}(p) - \hat{\mu}(q))$, where the empirical Kernel Mean Embeddings are $\hat{\mu}(p) = \sum_{i=1}^N a_i \Phi(x_i)$ and $\hat{\mu}(q) = \sum_{j=1}^n b_j \Phi(y_j)$. The empirical operator embeddings are given by $\hat{D}(q) = \sum_{j=1}^n b_j [J\Phi(y_j)]^\top J\Phi(y_j)$, and $\hat{C}_\gamma(q) = \sum_{j=1}^n b_j \Phi(y_j)\Phi(y_j)^\top - \gamma \hat{\mu}(q)\hat{\mu}(q)^\top$.

# 4   Unbalanced Continuous Kernel Sobolev Descent

Given the Kernel Sobolev-Fisher Discrepancy defined in the previous sections and its relation to advection-reaction transport, in this section we construct a Markov process that transports particles drawn from a source distribution to a target distribution. Note that we don't assume that the densities are normalized nor have same total mass.

## 4.1   Constructing the Continuous Markov Process

Given $\alpha, \lambda > 0, \gamma \in \{0,1\}$ and $n$ weighted particles drawn from the source distribution : $q^n_0 = q = \sum_{i=1}^n b_i \delta_{y_i}$, i.e $X^0_i = y_i$ and $w^0_i = b_i$. Recall that the target distribution is given by $p = \sum_{i=1}^N a_i \delta_{x_i}$. We define the following Markov Process that we name Unbalanced Kernel Sobolev Descent:

$$dX^i_t = \nabla_x u^{\lambda,\gamma}_{p,q^n_t}(X^i_t)dt \ \text{(advection step)}$$
$$dw^i_t = \alpha(u^{\lambda,\gamma}_{p,q^n_t}(X^i_t) - \gamma \mathbb{E}_{q^{(n)}_t} u^{\lambda,\gamma}_{p,q^n_t}(x))w^i_t dt \ \text{(reaction step)}$$
$$q^n_t = \sum_{i=1}^n w^i_t \delta_{X^i_t}, \tag{2}$$

where $u^{\lambda,\gamma}_{p,q^n_t}$ is the critic of the Kernel Sobolev-Fisher discrepancy, whose expression and gradients are given in Proposition 2. We see that USD consists of two steps: the advection step that updates the particles positions following the gradient flow of the Sobolev-Fisher critic, and a reaction step that updates the weights of the particles with a growth rate proportional to that critic. This reaction step consists in mass construction or destruction, that depends on the confidence of the witness function. This can be seen as birth-death process on the particles, where the survival $\log$ probability of a particle is proportional to the critic evaluation on this particle.

## 4.2   Generator Expression and PDE in the limit of $n \to \infty$

Proposition 3 gives the evolution equation of a functional of the intermediate distributions $q^n_t$ produced in the descent, at the limit of infinite particles $n \to \infty$:

**Proposition 3.** *Let* $\Psi : \mathcal{P}(\mathcal{X}) \to \mathbb{R}$, *be a functional on the probability space. Let* $q^n_t$ *be the distribution produced by USD at time t. Let* $q_t$ *be its limit as* $n \to \infty$, *we have:*

$$\partial_t \Psi[q_t] = (\mathcal{L}\Psi)[q_t],$$

where $\mathcal{L}\Psi(q) = \int \left\langle \nabla_x u_{p,q}^{\lambda;\gamma}(x), \nabla_x D_q \Psi(x) \right\rangle q(dx) + \alpha \int D_q \Psi(x)(u_{p,q}^{\lambda;\gamma}(x) - \gamma \mathbb{E}_q u_{p,q}^{\lambda;\gamma})q(x)dx$. Where the functional derivative $D_\mu$ is defined through first variation for a signed measure $\chi$ $(\int \chi(x)dx = 0)$:

$$\int D_\mu \Psi(x)\chi(x)dx = \lim_{\varepsilon \to 0} \frac{\Psi(\mu + \varepsilon\chi) - \Psi(\mu)}{\varepsilon}.$$

In particular, the paths of USD in the limit of $n \to \infty$ satisfy the advection-reaction equation:

$$\partial_t q_t = -div(q_t \nabla_x u_{p,q_t}^{\lambda,\gamma}) + \alpha(u_{p,q_t}^{\lambda,\gamma} - \gamma \mathbb{E}_{q_t} u_{p,q}^{\lambda,\gamma})q_t.$$

### 4.3 Unbalanced Sobolev Descent decreases the MMD.

The following Theorem shows that USD when the number of the particles goes to infinity decreases the MMD distance at each step, where: $\text{MMD}^2(p,q) = \|\mu(p) - \mu(q)\|^2$.

**Theorem 3** (Unbalanced Sobolev Descent decreases the MMD). *Consider the paths $q_t$ produced by USD. In the limit of particles $n \to \infty$ we have*

$$\frac{1}{2}\frac{dMMD^2(p,q_t)}{dt} = -\left(MMD^2(p,q_t) - \lambda SF_{\mathcal{H},\gamma,\lambda}^2(p,q_t)\right) \le 0. \tag{3}$$

In particular, in the regularized case $\lambda > 0$ with strict descent (i.e. $q_t \ne p$ implies $\text{MMD}^2(p,q_t) - \lambda SF_{\mathcal{H},\gamma,\lambda}^2(p,q_t) > 0$), USD converges in the MMD sense: $\lim_{t\to\infty} \text{MMD}^2(p,q_t) = 0$. Similarly to [2], strict descent is ensured if the kernel and the target distribution $p$ satisfy the condition: $\delta_{p,q} \notin \text{Null}(D(q) + \alpha C_\gamma(q)), \forall q \ne p$.

**USD Accelerates the Convergence.** We now prove a Lemma the can be used to show that Unbalanced Sobolev Descent has an acceleration advantage over Sobolev Descent [2].

**Lemma 1.** *In the regularized case $\lambda > 0$ with $\alpha > 0$, the Kernel Sobolev-Fisher Discrepancy $SF_{\mathcal{H},\gamma,\lambda}$ is strictly upper bounded by the Kernel Sobolev discrepancy $\mathcal{S}_{\mathcal{H},\lambda}$(i.e for $\alpha = 0$) [2]:*

$$SF_{\mathcal{H},\gamma,\lambda}^2(p,q) < \mathcal{S}_{\mathcal{H},\lambda}^2(p,q).$$

From Lemma 1 and Eq. (3), we see that USD ($\alpha > 0$), results in a larger decrease in MMD than SD [2] ($\alpha = 0$), resulting in a steeper descent. Hence, USD advantages over SD are twofold: 1) it allows unbalanced transport, 2) it accelerates convergence for the balanced and unbalanced transport.

**USD with Universal Infinite Dimensional Kernel.** While we presented USD with a finite dimensional kernel for ease of presentation, we show in Appendix D that all our results hold for an infinite dimensional kernel. For a universal or a characteristic kernel, convergence in MMD implies convergence in distribution (see [21, Theorem 12]). Hence, using a universal kernel, USD guarantees the weak convergence as $\text{MMD}(p,q_t) \to 0$.

### 4.4 Understanding the effect of the Reaction Step: Whitened Principal Transport Directions

In [2] it was shown that the gradient of the Sobolev Discrepancy can be written as a linear combination of principal transport directions of the Gramian of derivatives $D(q)$. Here we show that unbalanced descent leads to a similar interpretation in a whitened feature space thanks to the $\ell_2$ regularizer. Let $\tilde{\mathcal{H}}_q = \{f \ |f(x) = \left\langle v, \tilde{\Phi}_q(x) \right\rangle, \tilde{\Phi}_q(x) = (C_\gamma(q) + \frac{\lambda}{\alpha}I)^{-\frac{1}{2}}\Phi(x)\}$, $\tilde{\delta}_{p,q} = (C_\gamma(q) + \frac{\lambda}{\alpha}I)^{-\frac{1}{2}}\delta_{p,q}$ $\tilde{D}(q) = (C_\gamma(q) + \frac{\lambda}{\alpha}I)^{-\frac{1}{2}}D(q)(C_\gamma(q) + \frac{\lambda}{\alpha}I)^{-\frac{1}{2}}$, and let $v_{p,q}^{\lambda,\gamma} = (\tilde{D}(q) + \alpha I_m)^{-1}\tilde{\delta}_{p,q}$. It is easy to see that the critic of the SF can be written as: $u_{p,q}^{\lambda;\gamma}(x) = \left\langle u_{p,q}^{\lambda,\gamma}, \Phi(x) \right\rangle = \left\langle v_{p,q}^{\lambda,\gamma}, \tilde{\Phi}_q(x) \right\rangle$. Note that $\tilde{\Phi}_q$ is a whitened feature map and $\tilde{D}(q)$ is the Gramian of its derivatives. Let $\tilde{d}_j, \lambda_j$ be the eigenvectors and eigenvalues of $\tilde{D}(q)$. We have: $v_{p,q}^{\lambda,\gamma} = \sum_{j=1}^m \frac{1}{\lambda_j + \alpha}\tilde{d}_j \left\langle \tilde{d}_j, \tilde{\delta}_{p,q} \right\rangle$. Hence, we write the gradient of the Sobolev-Fisher critic as $\nabla_x u_{p,q}^{\lambda,\gamma}(x) = \sum_{j=1}^m \frac{1}{\lambda_j + \alpha}\left\langle \tilde{d}_j, \tilde{\delta}_{p,q} \right\rangle [J\tilde{\Phi}(x)]\tilde{d}_j = \sum_{j=1}^m \frac{1}{\lambda_j + \alpha}\left\langle \tilde{d}_j, \tilde{\delta}_{p,q} \right\rangle \nabla_x \tilde{d}_j(x)$, where $\tilde{d}_j(x) = \left\langle \tilde{d}_j, \tilde{\Phi}_q(x) \right\rangle$. This says that the mass is transported along a weighted combination of whitened principal transport directions $\nabla_x \tilde{d}_j(x)$. $\alpha$ introduces a damping of the transport as it acts as a spectral filter on the transport directions in the whitened space.

# 5 Discrete time Unbalanced Kernel and Neural Sobolev Descent

In order to get a practical algorithm in this Section we discretize the continuous USD given in Eq. (2). We also give an implementation parameterizing the critic as a Neural Network.

**Discrete Time Kernel USD.** Recall that the source distribution $q_0 = q = \sum_{j=1}^n b_j \delta_{y_j}$, note $w_j^0 = b_j$ and $x_j^0 = y_j, j = 1 \ldots n$. The target distribution $p = \sum_{j=1}^N a_j \delta_{x_j}$, and assume for simplicity $\sum_{j=1}^N a_j = 1$. Let $\varepsilon > 0$, for $\ell = 1 \ldots L$, for $j = 1 \ldots n$, we discretize the advection step:

$$x_j^\ell = x_j^{\ell-1} + \varepsilon \nabla_x u_{p,q_{\ell-1}}^{\lambda,\gamma}(x_j^{\ell-1}).$$

Let $m_{\ell-1} = \sum_{j=1}^n w_j^{\ell-1} u_{p,q_{\ell-1}}^{\lambda,\gamma}(x_j^{\ell-1})$. For $\tau > 0$, similarly we discretize the reaction step as:

$$a_j^\ell = \log(w_j^{\ell-1}) + \tau(u_{p,q_{\ell-1}}^{\lambda,\gamma}(x_j^{\ell-1}) - \gamma m_{\ell-1}).$$

If $\gamma = 0$ (total mass not conserved) we define the reweighing as follows: $w_j^\ell = \exp(a_j^\ell)$ and if $\gamma = 1$ (mass conserved): $w_j^\ell = \exp(a_j^\ell) / \sum_{i=1}^n \exp(a_i^\ell)$, and finally : $q^\ell = \sum_{j=1}^n w_j^\ell \delta_{x_j^\ell}$.

**Neural Unbalanced Sobolev Descent.** Motivated by the use of neural network critics in Sobolev Descent [2], we propose a Neural variant of USD by parameterizing the critic of the Sobolev-Fisher Discrepancy as a Neural network $f_\xi$ trained via gradient descent with the Augmented Lagrangian Method (ALM) on the loss function of SF given in Definition 1. The re-weighting is defined as in the kernel case above. Neural USD with re-weighting is summarized in Algorithm 1 in Appendix B. Note that the re-weighting can also be implemented via a birth-death process as in [12]. In this variant, particles are duplicated or killed with a probability driven by the growth rate given by the critic. We give the details of the implementation as birth-death process in Algorithm 2 (Appendix B).

**Computational and Sample Complexities.** The computational complexity Neural USD is given by that of updating the witness function and particles by SGD with backprop, i.e. $O(N(T + B))$, where $N$ is the mini-batch size, $T$ is the training time, $B$ is the gradient computation time for particles update. $T$ corresponds to a forward and a backward pass through the critic and its gradient. The sample complexity for estimating the Sobolev Fisher critic scales like $1/\sqrt{N}$ similar to MMD [22].

# 6 Relation to Previous Work

Table 1 in Appendix A summarizes the main differences between Sobolev descent [2], which only implements advection, and USD that also implements advection-reaction. Our work is related to the conic particle descent that appeared in [13] and [12]. The main difference of our approach is that it is not based on the flow of a fixed functional, but we rather learn dynamically the flow that corresponds to the witness function of the Sobolev-Fisher discrepancy. The accelerated Langevin Sampling of [14] also uses similar principles in the transport of distributions via Langevin diffusion and a reaction term implemented as a birth-death process. The main difference with our work is that in Langevin sampling the log likelihood of the target distribution is required explicitly, while in USD we only need access to samples from the target distribution. USD relates to unbalanced optimal transport [6, 7, 8, 9] and offers a computational flexibility when compared to Sinkhorn approaches [8, 9], since it scales linearly in the number of points while Sinkhorn is quadratic. Compared to WFR (Eq. (1)), USD finds greedily the connecting path, while WFR solves an optimal planning problem.

# 7 Applications

We experiment with USD on synthetic data, image coloring and prediction of developmental stages of scRNA-seq data. In all our experiments we report the MMD distance with a gaussian kernel, computed using the random Fourier features (RF) approximation [23] with 300 RF and kernel bandwith equal to $\sqrt{d}$ (the input dimension). We consider the conservation of mass case, i.e. $\gamma = 1$.

**Synthetic Examples.** We test Neural USD descent (Algorithms 1 and 2) on two synthetic examples. In the first example (Figure 1), the source samples are drawn from a 2D standard Gaussian, while target samples are drawn from a Mixture of Gaussians (MOG). Samples from this MOG have uniform

weights. In the second example (Figure 2), source samples are drawn from a 'cat'-shaped density whereas the target samples are drawn uniformly from a 'heart'. Samples from the targets have non-uniform weights following a horizontal gradient. In order to target such complex densities USD exploits advection and reaction by following the critic gradients and by creation and destruction of mass. We see in Figs 1 and 2 a faster mixing of USD in both, implementation with weights **(w)** and as birth-death **(bd)** processes compared to the Sobolev descent algorithm of [2].

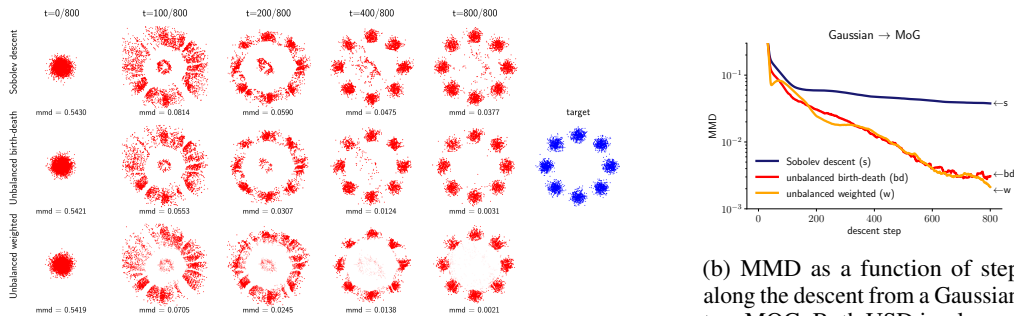

(a) Neural USD paths in transporting a Gaussian to a MOG. We compare Sobolev descent (SD, [2]) to both USD implementations: with birth-death process (bd: Algorithm 2) and weights (w: Algorithm 1). USD outperforms SD in capturing the modes of the MOG.

(b) MMD as a function of step along the descent from a Gaussian to a MOG. Both USD implementations convergence faster to the target distribution, reaching lower MMD than Sobolev Descent that relies on advection only.

Figure 1: Neural USD transport of a Gaussian to a MOG (target distribution is uniformly weighted).

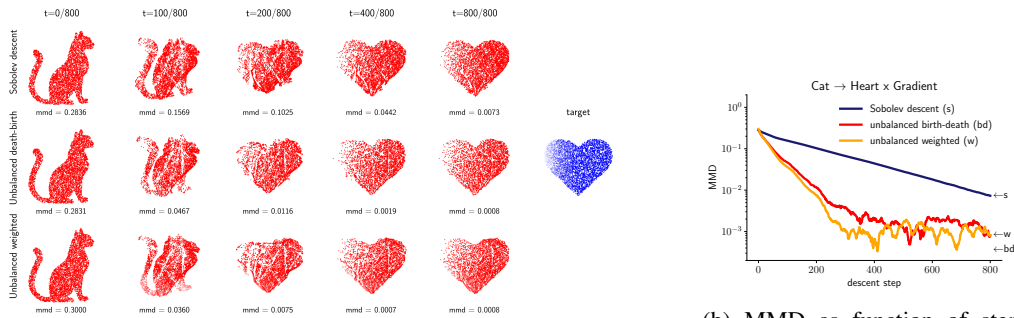

(a) Neural USD transporting a 'cat' distributed cloud to a 'heart'. The main difference with the example above is that the points of the target distribution have non uniform weights describing a linear gradient as seen from the color code in the figure. Similarly to the MOG case, USD outperforms SD and better captures the non uniform density of the target.

(b) MMD as function of step along the descent from cat → heart × Grad. Similarly to the uniform target case USD accelerates the descent and outperforms SD.

Figure 2: Neural USD transport of a 'cat' to a non-uniform 'heart'. Samples from the target distribution have non-uniform weights given by $a_j$'s following a linearly decaying gradient.

**Image Color Transfer.** We test Neural USD on the image color transfer task. We choose target images that have sparse color distributions. This is a good test for unbalanced transport since intuitively having birth and death of particles accelerates the transport convergence in this case. We compare USD to standard optimal transport algorithms. We follow the recipe of [24] as implemented in the POT library [25], where images are subsampled for computational feasibility and then interpolated for out-of-sample points. We compare USD to Earth-Moving Distance (EMD), Sinkhorn [26] and Unbalanced Sinkhorn [8] baselines. We see in Figure 3 that USD achieves smaller MMD to the target color distribution. We give in Appendix H.2 in Fig 7 trajectories of the USD.

**Developmental Trajectories of Single Cells.** When the goal is not only to transport particles but also to find intermediate points along trajectories, USD becomes particularly interesting. This type of use case has recently received increased attention in developmental biology, thanks to single-cell RNA sequencing (scRNA-seq), a technique that records the expression profile of a whole population of cells at a given stage, but does so destructively. In order to trace the development of cells in-between such destructive measurements, [16] proposed to use unbalanced optimal transport [8].

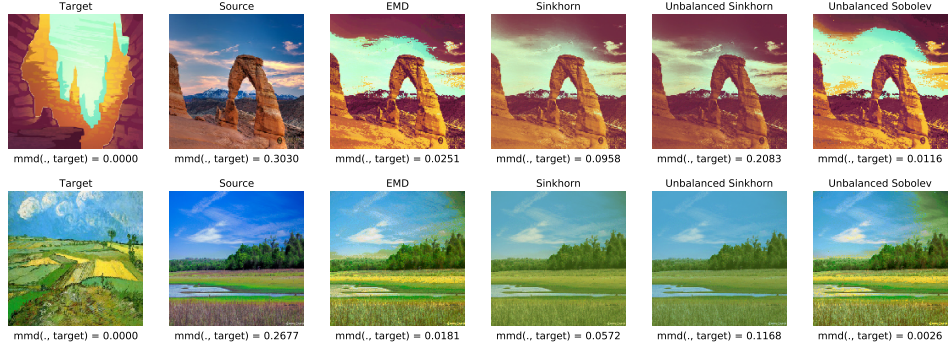

| Target | Source | EMD | Sinkhorn | Unbalanced Sinkhorn | Unbalanced Sobolev |
|---|---|---|---|---|---|
| mmd(., target) = 0.0000 | mmd(., target) = 0.3030 | mmd(., target) = 0.0251 | mmd(., target) = 0.0958 | mmd(., target) = 0.2083 | mmd(., target) = 0.0116 |
| mmd(., target) = 0.0000 | mmd(., target) = 0.2677 | mmd(., target) = 0.0181 | mmd(., target) = 0.0572 | mmd(., target) = 0.1168 | mmd(., target) = 0.0026 |

Figure 3: Color Transfer with USD using (bd) Algorithm 2. Comparison to OT baselines (EMD, Sinkhorn and Unbalanced Sinkhorn). USD achieves lower MMD, and faithfully captures the sparse distribution of the target.

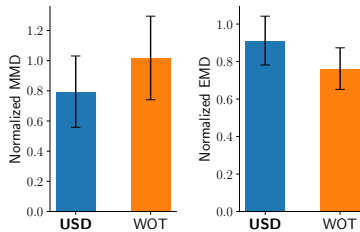

Figure 4: Mean and standard deviations plots of Normalized MMD and EMD for the intermediate stage prediction by USD and WOT (unbalanced OT) of [16] (means and standards deviation are computed over intervals). While USD outperforms WOT in MMD, the reverse holds in EMD. See text for an explanation.

Denoting those populations $q_{t_0}$ (source) and $q_{t_1}$ (target), then, in order to predict the population at an intermediate time $\frac{t_0+t_1}{2}$, [16] used a linear interpolation between matches between the source and target populations based on the coupling of unbalanced OT. This type of interpolation is a form of McCann interpolate [27]. As an alternative, we propose to use the mid-point of the USD descent as an interpolate, i.e. the timestamp in the descent $t_{1/2}$ such that $\text{MMD}(q_{t_{1/2}}, q_{t_0}) = \text{MMD}(q_{t_{1/2}}, q_{t_1})$. We test this procedure on the dataset released by [16]. For all time intervals $[t_0, t_1]$ in the dataset, we compute the intermediate stage $q_{t_{1/2}}$. We compare the quality of this interpolate with that obtained by the WOT algorithm of [16] in terms of MMD to the ground truth intermediate population $q_{t_{1/2}}^*$, normalized by MMD between initial and final population, i.e. $\text{MMD}(q_{t_{1/2}}, q_{t_{1/2}}^*)/\text{MMD}(q_{t_0}, q_{t_1})$. Fig. 4 gives mean and standard deviation of the normalized MMD between intermediate stages predicted by USD and the ground truth. Note that mean and standard deviations are computed across 35 time intervals, individual MMDs can be found in Figure 8 in Appendix H. From Figure 4 we see that USD outperforms WOT in MMD, since USD is designed to decrease the MMD distance. On the other hand, for fairness of the evaluation we also report Normalized EMD (Earth-Mover Distance, normalized similarly) for which WOT outperforms USD. This is not surprising since WOT relies on unbalance OT, while USD instead provides guarantees in terms of MMD.

## 8 Conclusion

In this paper we introduced the KSFD discrepancy and showed how it relates to an advection-reaction transport. Using the critic of KSFD, we introduced Unbalanced Sobolev Descent (USD) that consists in an advection step that moves particles and a reaction step that re-weights their mass. The reaction step can be seen as birth-death process which, as we show theoretically, speeds up the descent compared to previous particle descent algorithms. We showed that the MMD convergence of Kernel USD and presented two neural implementations of USD, using weight updates, and birth and death of particle, respectively. We empirically demonstrated on synthetic examples and in image color transfer, that USD can be reliably used in transporting distributions, and indeed does so with accelerated convergence, supporting our theoretical analysis. As a further demonstration of our algorithm, we showed that USD can be used to predict developmental trajectories of single cells based on their RNA expression profile. This task is representative of a situation where distributions of different mass need to be compared and interpolated between, since the different scRNA-seq measurements are taken on cell populations of dissimilar size at different developmental stages. USD can naturally deal with this unbalanced setting. Finally we compared USD to unbalanced OT algorithms, showing its viability as a data-driven, more scalable dynamic transport method.

## Broader Impact Statement

Our work provides a practical particle descent algorithm that comes with a formal convergence proof and theoretically guaranteed acceleration over previous competing algorithms. Moreover, our algorithm can naturally handle situations where the objects of the descent are particles sampled from a source distribution descending towards a target distribution with different mass.

The type of applications that this enables range from theoretically principled modeling of biological growths processes (like tumor growth) and developmental processes (like the differentiation of cells in their gene expression space), to faster numerical simulation of advection-reaction systems.

Since our advance is mainly theoretical and algorithmic (besides the empirical demonstrations), its implications are necessarily tied to the utilization for which it is being deployed. Beside the applications that we mentioned, particle descent algorithms like ours have been proposed as a paradigm to characterize and study the dynamics of Generative Adversarial Network (GANs) training. As such, they could indirectly contribute to the risks associated with the nefarious uses of GANs such as deepfakes. On the other hand, by providing a tools to possibly analyze and better understand GANs, our theoretical results might serve as the basis for mitigating their abuse.

## Acknowledgments and Disclosure of Funding

Authors did not receive any third party funding and have no competing interests.

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
