[Supplementary Material]

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

## .1 Relation to Unbalanced Optimal Transport

We now relate our definition of the Sobolev-Fisher discrepancy to the following norm. For a signed measure $\chi$ define $\|\chi\|^2_{\dot{H}^{-1,2}(\nu)} =$

$$\sup_{f, \int_{\mathcal{X}}(\|\nabla_x f(x)\|^2 + \alpha f^2(x))d\nu \leq 1} \left| \int f d\chi \right| = \inf_{f, \chi(x) = -div(\nu(x)\nabla_x f(x)) + \alpha f(x)\nu(x)} \int_{\mathcal{X}}(\|\nabla_x f\|^2 + \alpha f^2)d\nu.$$

It can be shown that $\mathrm{SF}^2(p, q) = \|p - q\|^2_{\dot{H}^{-1,2}(q)}$.

The dynamic formulation of the Wasserstein Fisher-Rao metric given in Equation (1) can therefore be compactly written as:

$$\mathrm{WFR}^2(p, q) = \inf_{\nu_t} \int_0^1 \|d\nu_t\|^2_{\dot{H}^{-1,2}(\nu_t)}. \tag{4}$$

From this connection to WFR through $\|.\|^2_{\dot{H}^{-1,2}(q)}$, we see the link of the Sobolev-Fisher discrepancy to unbalanced optimal transport, since it linearizes the WFR for small perturbations.

## A Summary Table

| | $\alpha$ | $\gamma$ | Markov Process Particles $j = 1 \ldots n$ | PDE (As $n \to \infty$) | Guarantee $\frac{1}{2}\frac{d\mathrm{MMD}^2(p,q_t)}{dt} =$ |
|---|---|---|---|---|---|
| **Sobolev Descent** Flow of $\mathcal{S}_{\mathcal{H},\lambda}$ Target: $p = \frac{1}{N}\sum_{i=1}^N \delta_{x_i}$ Source : $q = \frac{1}{n}\sum_{j=1}^n \delta_{y_j}$ | 0 | N/A | $dX_t^j = \nabla_x u_{q_t}^\lambda(X_t^j)dt$ $q_t = \frac{1}{n}\sum_{j=1}^n \delta_{X_t^j}$ Principal Transport Directions: $dX_t = \sum_{\ell=1}^m \frac{1}{\lambda_\ell + \lambda}\langle \tilde{d}_\ell, \delta_{p,q_t}\rangle \nabla_x d_\ell(x)dt$ $(\lambda_j, d_j) = eig(D(q_t))$ | $\partial_t q_t = -div(q_t \nabla_x u_{p,q_t}^\lambda)$ Advection | $-(\mathrm{MMD}^2(p, q_t) - \lambda\mathcal{S}_{\mathcal{H},\lambda}^2(p, q_t))$ |
| **Unbalanced Sobolev Descent:** Flow of $\mathrm{SF}_{\mathcal{H},\lambda}$ Target: $p = \sum_{i=1}^N a_i\delta_{x_i}$ Source : $q = \sum_{j=1}^n b_j\delta_{y_j}$ $(\sum_i a_i \neq \sum_j b_j)$ | $\alpha > 0$ | $\gamma = 0$ | $dX_t^j = \nabla_x u_{p,q_t}^{\lambda,\gamma}(X_t^j)dt$ $dw_t^j = \alpha(u_{p,q_t}^{\lambda,\gamma}(X_t^j))w_t^j dt$ $q_t = \sum_{i=1}^n w_t^i\delta_{X_t^i}$ Whitened Principal Transport Directions : $dX_t^j = \sum_{\ell=1}^m \frac{1}{\lambda_\ell + \alpha}\langle \tilde{d}_\ell, \tilde{\delta}_{p,q}\rangle\nabla_x\tilde{d}_\ell(X_t^i)dt$ | $\partial_t q_t = -div(q_t\nabla_x u_{p,q_t}^{\lambda,\gamma}) + \alpha u_{p,q_t}^{\lambda,\gamma}(x)q_t$ Advection/Reaction (Mass not conserved ) | $-(\mathrm{MMD}^2(p, q_t) - \lambda\mathrm{SF}_{\mathcal{H}}^2(p, q_t))$ |
| **Balanced Sobolev Descent:** Flow of $\overline{\mathrm{SF}}_{\mathcal{H},\lambda}$ Target: $p = \sum_{i=1}^N a_i\delta_{x_i}$ Source : $q = \sum_{j=1}^n b_j\delta_{y_j}$ $(\sum_i a_i = \sum_j b_j)$ | $\alpha > 0$ | $\gamma = 1$ | $dX_t^j = \nabla_x u_{p,q_t}^{\lambda,\gamma}(X_t^j)dt$ $dw_t^j = \alpha(u_{p,q_t}^{\lambda,\gamma}(X_t^j) - \mathbb{E}_{q_t}u_{p,q_t}^{\lambda,\gamma})w_t^j dt$ $q_t = \sum_{i=1}^n w_t^i\delta_{X_t^i}$ Whitened Principal Transport Directions : $dX_t^j = \sum_{\ell=1}^m \frac{1}{\lambda_\ell + \alpha}\langle \tilde{d}_\ell, \tilde{\delta}_{p,q}\rangle\nabla_x\tilde{d}_\ell(X_t^i)dt$ | $\partial_t q_t = -div(q_t\nabla_x u_{p,q_t}^{\lambda,\gamma}) + \alpha(u_{p,q_t}^{\lambda,\gamma}(x) - \mathbb{E}_{q_t}u_{p,q_t}^{\lambda,\gamma})q_t$ Advection/Reaction (Mass conserved ) | $-(\mathrm{MMD}^2(p, q_t) - \lambda\overline{\mathrm{SF}}_{\mathcal{H}}^2(p, q_t))$ |

Table 1: Summary table comparing Unbalanced Sobolev Descent to Sobolev Descent.

## B Algorithms

---

**Algorithm 3** CRITIC UPDATE($\xi$, target $\{(a_i, x_i)\}$, current source $\{(w_j^{\ell-1}, x_j^{\ell-1})\}$,$\gamma$)

---

**for** $j = 1$ **to** $n_c$ **do**
   $m_\xi \leftarrow \sum_{j=1}^n w_j^{\ell-1} f_\xi(x_j^{\ell-1})$
   $\hat{\mathcal{E}}(\xi) \leftarrow \sum_{i=1}^N a_i f_\xi(x_i) - m_\xi$
   $\hat{\Omega}(\xi) \leftarrow \sum_j w_j^{\ell-1}\|\nabla_x f_\xi(x_j^{\ell-1})\|^2 + \alpha\left(\sum_j w_j^{\ell-1}f_\xi^2(x_j^{\ell-1}) - \gamma m_\xi^2\right)$
   $\mathcal{L}_S(\xi, \lambda) = \hat{\mathcal{E}}(\xi) + \lambda(1 - \hat{\Omega}(\xi)) - \frac{\rho}{2}(\hat{\Omega}(\xi) - 1)^2$
   $(g_\xi, g_\lambda) \leftarrow (\nabla_\xi\mathcal{L}_S, \nabla_\lambda\mathcal{L}_S)(\xi, \lambda)$
   $\xi \leftarrow \xi + \eta \, \mathrm{ADAM}\,(\xi, g_\xi)$
   $\lambda \leftarrow \lambda - \rho g_\lambda$ {SGD rule on $\lambda$ with learning rate $\rho$}
**end for**
**Output:** $\xi$

---

---

**Algorithm 1** **w**-Neural Unbalanced Sobolev Descent (weighted version – ALM Algorithm)

---

**Inputs:** $\varepsilon, \tau$ Learning rate particles, $n_c$ number of critics updates, $L$ number of iterations, $\gamma \in \{0, 1\}$
$\{(a_i, x_i), i = 1 \ldots N\}$, drawn from target distribution $\nu_p$
$\{(b_j, y_j), j = 1 \ldots n\}$ drawn from source distribution $\nu_q$
Neural critic $f_\xi(x) = \langle v, \Phi_\omega(x) \rangle, \xi = (v, \omega)$ parameters of the neural network
**Initialize** $x_j^0 = y_j, w_j^0 = b_j$ for $j = 1 \ldots n$
**for** $\ell = 1 \ldots L$ **do**
  *Critic Parameters Update*
  (between particles updates, gradient descent on the critic is initialized from previous episodes)
  $\xi \leftarrow$ CRITIC UPDATE($\xi$, target $\{x_i\}$, current source $\{(w_j^{\ell-1}, x_j^{\ell-1})\}, \gamma$ ) (Given in Alg. 3 in Appendix B)
  *Particles and Weights Update*
  **for** $j = 1$ **to** $n$ **do**
    $x_j^\ell = x_j^{\ell-1} + \varepsilon \nabla_x f_\xi(x_j^{\ell-1})$ (current $f_\xi$ is the critic between $q_{\ell-1}$ and $p$, advection step)
    $a_j^\ell = \log(w_j^{\ell-1}) + \tau(f_\xi(x_j^{\ell-1}) - \gamma m_\xi)$ (reaction step)
    **if** $\gamma = 1$ (mass conservation) **then**
      $w^\ell = \text{Softmax}(a^\ell) \in \Delta_n$
    **else if** $\gamma = 0$ (mass not conserved) **then**
      $w^\ell = \exp(a^\ell)$
    **end if**
  **end for**
**end for**
**Output:** $\{(x_j^L, w_j^L), j = 1 \ldots n\}$

---

---

**Algorithm 2** **bd**-Neural Unbalanced Sobolev Descent (Birth-Death – ALM Algorithm)

---

**Inputs:** Same inputs of Algorithm 1
**Initialize** $x_j^0 = y_j, w_j^0 = \frac{1}{n}$ for $j = 1 \ldots n$
**for** $\ell = 1 \ldots L$ **do**
  *Critic Parameters Update*
  (between particles updates gradient descent on the critic is initialized from previous episodes)
  $\xi \leftarrow$ CRITIC UPDATE($\xi$, target $\{(a_i, x_i)\}$, current source $\{(\frac{1}{n}, x_j^{\ell-1})\}, \gamma$ ) (Given in Alg. 3 in App. B)
  *Particles and Weights Update (birth-death)*
  **for** $j = 1$ **to** $n$ **do**
    $x_j^\ell = x_j^{\ell-1} + \varepsilon \nabla_x f_\xi(x_j^{\ell-1})$ (current $f_\xi$ is the critic between $q_{\ell-1}$ and $p$ )
    $m_\xi \leftarrow \frac{1}{n} \sum_{i=1}^{j} f_\xi(x_i^\ell) + \frac{1}{n} \sum_{i=j+1}^{n} f_\xi(x_i^{\ell-1})$
    **if** $\beta_j = f_\xi(x_j^\ell) - \gamma m_\xi > 0$ **then**
      Duplicate $x_j^\ell$ with probability $1 - \exp(-\alpha \tau \beta_j)$
    **else if** $\beta_j = f_\xi(x_j^\ell) - \gamma m_\xi < 0$ **then**
      kill $x_j^\ell$ with probability $1 - \exp(-\alpha \tau |\beta_j|)$
    **end if**
  **end for**{Make population size $n$ again}
  $n_\ell$ number of particles at the end of the loop
  **if** $n_\ell > n$ **then**
    Kill $n_\ell - n$ randomly selected particles
  **else if** $n_\ell < n$ **then**
    Duplicate $n - n_\ell$ randomly selected partciles
  **end if**
**end for**
**Output:** $\{(x_j^L), j = 1 \ldots n\}$

---

## C  Proofs

*Proof of Theorem 1.* Define the following dot product between $u, v$ in the the Sobolev Space:

$$\langle u, v\rangle_{W_0^2} = \int_{\mathcal{X}} \langle \nabla_x u(x), \nabla_x v(x)\rangle\, q(x) + \alpha \int_{\mathcal{X}} u(x)v(x)q(x)dx,$$

and the norm :

$$\|u\|_{W_0^2}^2 = \int_{\mathcal{X}} \|\nabla_x u(x)\|^2 q(x)dx + \alpha \int_{\mathcal{X}} u^2(x)q(x)dx,$$

Let $f$ be any function such that $f|_{\partial \mathcal{X}=0}$, and $\|f\|_{W_0^2} \leq 1$:

$$\begin{aligned}
\mathcal{E}(f) &= \int_{\mathcal{X}} f(x)(p(x) - q(x))dx \\
&= -\int_{\mathcal{X}} f(x)div(q(x)\nabla_x u(x))dx + \alpha \int_{\mathcal{X}} u(x)f(x)q(x) \\
&= \int_{\mathcal{X}} \langle \nabla_x f(x), \nabla_x u(x)\rangle\, q(x) + \alpha \int_{\mathcal{X}} u(x)f(x)q(x)dx \\
&= \langle u, f\rangle_{W_0^2} \text{ (By definition)} \\
&\leq \|u\|_{W_0^2} \|f\|_{W_0^2} \text{ (By Cauchy Schwarz) ,} \\
&\leq \|u\|_{W_0^2} \text{ ($f$ feasible, $\|f\|_{W_0^2} \leq 1$)}
\end{aligned}$$

Let $f_{p,q}^* = u/\|u\|_{W_0^2}$, we have $\|f_{p,q}^*\|_{W_0^2} = 1$ and hence feasible, and it is easy to see that :

$$\mathcal{E}(f_{p,q}^*) = \|u\|_{W_0^2},$$

and hence we have that for all $f$ feasible we have:

$$\mathcal{E}(f) \leq \mathcal{E}(f_{p,q}^*),$$

and hence $f_{p,q}^*$ achieves the sup. $\qquad \square$

*Proof of Proposition 1.* This can be easily proved using that $u^*$ solution of the PDE with source term is solution of that sup problem. $L(u^*) = \text{SF}^2(p, q)$ is clear from definition of $u^*$ we are left showing $L(u) \leq L(u*)$ for all $u$, this can be shown by proving that :

$$L(u) - L(u^*) = -\|u - u^*\|_{W_0^2}^2 \leq 0$$

and hence $L(u) \leq L(u^*)$ , hence $u^*$ achieves the sup. $\qquad \square$

*Proof of Theorem 2.* Writing the Lagrangian $u$ we have:

$$\inf_{V,r} \sup_u \mathcal{L}(V, r, u) = \sup_u \inf_{V,r} \mathcal{L}(V, r, u),$$

where By convexity of the cost we exchange sup and inf for $\mathcal{L}(V, r, u) = \frac{1}{2}\int_{\mathcal{X}} \|V(x)\|^2 q(x)dx + \alpha\frac{1}{2}\int_{\mathcal{X}} r^2(x)q(x)dx + \int_{\mathcal{X}} u(x)(p(x) - q(x)) - \int_{\mathcal{X}} \langle \nabla_x u(x), V(x)\rangle\, q(x) - \alpha \int_{\mathcal{X}} r(x)u(x)q(x)$.
Note that $\inf_V \int_{\mathcal{X}} \|V(x)\|^2 q(x)dx - \int_{\mathcal{X}} \langle \nabla_x u(x), V(x)\rangle\, q(x) = -\sup_V \int_{\mathcal{X}} \langle \nabla_x u(x), V(x)\rangle\, q(x) - \frac{1}{2}\int_{\mathcal{X}} \|V(x)\|^2 q(x)dx = -\frac{1}{2}\int_{\mathcal{X}} \|\nabla_x u(x)\|^2 q(x)dx$(Fenchel Convex).
Similarly we have: $\inf_r \frac{1}{2}\int_{\mathcal{X}} r^2(x)q(x)dx - \int_{\mathcal{X}} r(x)u(x)q(x) = -\sup_r \int_{\mathcal{X}} r(x)u(x)q(x) - \frac{1}{2}\int_{\mathcal{X}} r^2(x)q(x)dx = -\frac{1}{2}\int_{\mathcal{X}} u^2(x)q(x)dx$. Hence the dual problem is :

$$P = \sup_u \int_{\mathcal{X}} u(x)(p(x) - q(x))dx - \frac{1}{2}\left(\int_{\mathcal{X}} \|\nabla_x u(x)\|^2 q(x)dx + \alpha \int_{\mathcal{X}} u^2(x)q(x)dx\right)$$

By Proposition 1 ,we have :

$$P = \frac{1}{2}\text{SF}^2(p, q)$$

Hence $\text{SF}^2(p, q)$ has the equivalent form :

$$\text{SF}^2(p, q) = \inf_{V,r} \int_{\mathcal{X}} \|V(x)\|^2 q(x)dx + \alpha \int_{\mathcal{X}} r^2(x)q(x)dx$$

Subject to: $p(x) - q(x) = -div(q(x)V(x)) + \alpha r(x)q(x)$

Since $V^* = \nabla_x u$ and $r^* = u$ we have finally:

$$\text{SF}^2(p, q) = \inf_{u} \int_{\mathcal{X}} \|\nabla_x u(x)\|^2 q(x)dx + \alpha \int_{\mathcal{X}} u^2(x)q(x)dx$$

Subject to: $p(x) - q(x) = -div(q(x)\nabla_x u(x)) + \alpha u(x)q(x)$.

$\square$

*Proof of Proposition 2.*

$$
\begin{aligned}
L_{\gamma,\lambda}(u) &= 2(\mathbb{E}_{x\sim p}u(x) - \mathbb{E}_{x\sim q}u(x)) - \left(\mathbb{E}_{x\sim q}[\|\nabla_x u(x)\|^2 + \alpha(u(x) - \gamma\mathbb{E}_q u(x))^2] + \lambda\|u\|_{\mathcal{H}}^2\right) \\
&= 2\langle u, \mu(p) - \mu(q)\rangle_{\mathcal{H}} - \left(\langle u, D(q)u\rangle_{\mathcal{H}} + \alpha(\mathbb{E}_q u^2(x) - \gamma(\mathbb{E}_{x\sim q}u(x))^2) + \lambda\|u\|_{\mathcal{H}}^2\right) \\
&= 2\langle u, \mu(p) - \mu(q)\rangle_{\mathcal{H}} - \left(\langle u, D(q)u\rangle_{\mathcal{H}} + \alpha(\langle u, C(q)u\rangle_{\mathcal{H}} - \gamma(\langle u, \mu(q)\rangle_{\mathcal{H}})^2 + \lambda\|u\|_{\mathcal{H}}^2\right) \\
&= 2\langle u, \mu(p) - \mu(q)\rangle_{\mathcal{H}} - \langle u, (D(q) + \alpha(C(q) - \gamma\mu(q) \otimes \mu(q)) + \lambda I)u\rangle_{\mathcal{H}}
\end{aligned}
$$

Setting first order optimality for the sup we obtain:

$$(D(q) + \alpha(C(q) - \gamma\mu(q) \otimes \mu(q)) + \lambda I)\, u_{p,q}^{\lambda;\gamma} = \mu(p) - \mu(q) = \delta_{p,q}.$$

$\square$

*Proof of Proposition 3.* For simplicity we give here the proof for $\gamma = 1$. $\gamma = 0$ has a similar proof. The proof follows ideas from [12]. Let $\Psi$ be a measure valued functional $\Psi : \mathcal{P}(\mathbb{R}^d) \to \mathbb{R}$. For a measure $\mu$, $\Psi(\mu) \in \mathbb{R}$. The functional derivative $D_\mu$ is defined through first variation for a signed measure $\chi$ ($\int \chi(x)dx = 0$):

$$\int D_\mu \Psi(x)\chi(x)dx = \lim_{\varepsilon \to 0} \frac{\Psi(\mu + \varepsilon\chi) - \Psi(\mu)}{\varepsilon}$$

A generator function is defined as follows for a measure valued markov process $\mu_t^{(n)}$ (defined with $n$ particles) is defined as follows:

$$(\mathcal{L}_n \Psi)[\mu^{(n)}] = \lim_{s \to 0^+} \frac{\mathbb{E}_{\mu_0^n = \mu^{(n)}}(\Psi[\mu_s^{(n)}]) - \Psi(\mu^{(n)})}{s}$$

where

$$\mathbb{E}_{\mu_0^n = \mu^{(n)}}(\Psi[\mu_s^{(n)}]),$$

is the expectation of the functional $\Psi$ evaluated on the trajectory of the markov process $\mu_s^{(n)}$ taken on conditional on the initial step $\mu_0^{(n)} = \mu^{(n)}$.

1. Given our markov process i.e $\mu_t^{(n)}$ and $\mu_0^{(n)}$ we find the expression of the generator $\mathcal{L}_n \Psi[\mu^{(n)}]$ (using pertrubation analysis )

2. Since the process is markovian letting $t \to 0$ and considering the generator it will give us the evolution also between $t$ and $t + dt$ of $\Psi[\mu_t^{(n)}]$:

$$\partial_t \Psi(\mu_t^{(n)}) = (\mathcal{L}_n \Psi)[\mu_t^{(n)}], \Psi(\mu_t^{(n)})|_{t=0} = \Psi(\mu_0^{(n)})$$

3. Consider $n \to \infty$, identify the PDE corresponding to the generator

As $s \to 0$, and $\varepsilon \to 0$, we have:

$$E_0 \Psi(q_s^n) - q^n = \underbrace{E_0 \Psi(q_s^n) - E_0 q_{s-\varepsilon}^n}_{\text{weights updates}} + \underbrace{E_0 q_{s-\varepsilon}^n - q^n}_{\text{advection}}$$

The advection part:

$$A_n \Psi[q^n] = \sum_{j=1}^{n} w_j \int \left\langle \nabla_x u_{p,q^{(n)}}(X^j) \delta_{X_j}(dx), \nabla_x D_{q^n} \Psi(X^j) \right\rangle$$

$$= \int \left\langle \nabla_x u_{p,q^n}(x), \nabla_x D_{q^n} \Psi(x) \right\rangle) q^n(dx)$$

For the weight update part note that we have:

$$w_s^j = w_{s-\varepsilon}^j + \varepsilon \alpha (u_{p,q_{s-\varepsilon}^n}(X_{s-\varepsilon}^j) - \mathbb{E}_{q_{s-\varepsilon}^{(n)}} u_{p,q_{s-\varepsilon}^n}) w_{s-\varepsilon}^j$$

$$q_s^n = \sum_{j=1}^{N} w_s^j \delta_{X_{s-\varepsilon}^j}$$

$$q_s^n = q_{s-\varepsilon}^n + \varepsilon' \alpha \sum_{j=1}^{n} w_{s-\varepsilon}^j (u_{p,q_{s-\varepsilon}^n}(X_{s-\varepsilon}^j) - \mathbb{E}_{q_{s-\varepsilon}^{(n)}} u_{p,q_{s-\varepsilon}^n}) \delta_{X_{s-\varepsilon}^j}$$

Hence we have:

$$\frac{q_s^n(x) - q_{s-\varepsilon}^n(x)}{\varepsilon'} = \alpha (u_{p,q_{s-\varepsilon}^n}(x) - \mathbb{E}_{q_{s-\varepsilon}^{(n)}} u_{p,q_{s-\varepsilon}^n}) q_{s-\varepsilon}^n(x) = \chi$$

Hence the variation of $\Phi$:

$$\lim_{\varepsilon' \to 0} \frac{\Psi(q_s^n) - \Psi(q_{s-\varepsilon}^n)}{\varepsilon'} = \int D_{q_{s-\varepsilon}^n} \Psi(x) d\chi(x) = \alpha \int D_{q_{s-\varepsilon}^n} \Psi(x) (u_{p,q_{s-\varepsilon}^n}(x) - \mathbb{E}_{q_{s-\varepsilon}^{(n)}} u_{p,q_{s-\varepsilon}^n}) q_{s-\varepsilon}^n(x) dx$$

As $s, \varepsilon \to 0$ we obtain the effect of weights updates as follows:

$$W_n \Psi[q^n] = \alpha \int D_{q^n} \Psi(x) (u_{p,q^n}(x) - \mathbb{E}_{q^{(n)}} u_{p,q^n}) q^n(x) dx$$

Hence the Generator has the following form:

$$(\mathcal{L}_n \Psi)[q^{(n)}] = \int \left\langle \nabla_x u_{p,q^n}(x), \nabla_x D_{q^n} \Psi(x) \right\rangle) q^n(dx) + \alpha \int D_{q^n} \Psi(x) (u_{p,q^n}(x) - \mathbb{E}_{q^{(n)}} u_{p,q^n}) q^n(x) dx$$

and we have:

$$\partial_t \Psi[q_t^n] = (\mathcal{L}_n \Psi)[q_t^{(n)}], with q_0^{(n)} = q$$

As $n \to \infty$ we have the evolution of the PDE:

$$\partial_t q_t = -div(q(x) \nabla_x u_{p,q_t}) + \alpha (u_{p,q_t} - \mathbb{E}_{q_t} u_{p,q_t})$$

and

$$\partial_t \Psi[q_t] = (\mathcal{L}\Psi)[q_t],$$

where $\mathcal{L}\Psi(q) = \int \langle \nabla_x u_{p,q}(x), \nabla_x D_q \Psi(x) \rangle q(dx) + \alpha \int D_q \Psi(x) (u_{p,q}(x) - \mathbb{E}_q u_{p,q}) q(x) dx$.

$\square$

*Proof of Theorem 3 (Decrease of the MMD loss of the (Continous) Gradient Flow).* For $u_{p,q_t}^{\gamma,\lambda}$ we omit the up-scripts $\gamma$ and $\lambda$ in the following. Note that we have the following two expressions using the fact our functions are in the RKHS:

$$\int \langle \nabla_x u_{p,q_t}(x), \nabla_x \delta_{p,q_t} \rangle q_t(dx) = \int \left\langle u_{p,q_t}, (J\Phi(x))^\top J\Phi(x) \delta_{p,q_t} \right\rangle q_t(dx)$$

$$= \left\langle u_{p,q_t}, \mathbb{E}_{q_t} (J\Phi(x))^\top (J\Phi(x)) \delta_{p,q_t} \right\rangle$$

$$= \left\langle u_{p,q_t}, D(q_t) \delta_{p,q_t} \right\rangle.$$

On the other hand:

$$\int \delta_{p,q_t}(x)(u_{p,q_t}(x) - \gamma \mathbb{E}_{q_t} u_{p,q_t}) q_t(x) dx$$

$$= \int \langle \delta_{p,q_t}, \Phi(x) \rangle \langle \Phi(x) - \gamma \mu(q_t), u_{p,q_t} \rangle q_t(x) dx$$

$$= \int \langle \delta_{p,q_t}, \Phi(x) - \gamma \mu(q_t) \rangle \langle \Phi(x) - \gamma \mu(q_t), u_{p,q_t} \rangle q_t(x) dx$$

$$+ \quad \gamma \int \langle \delta_{p,q_t}, \mu(q_t) \rangle \langle \Phi(x) - \gamma \mu(q_t), u_{p,q_t} \rangle q_t(x) dx$$

$$= \quad \left\langle \delta_{p,q_t}, \left( \int (\Phi(x) - \gamma \mu(q_t)) \otimes (\Phi(x) - \gamma \mu(q_t)) q_t(dx) \right) u_{p,q_t} \right\rangle$$

$$+ \quad \gamma \langle \delta_{p,q_t}, \mu(q_t) \rangle \int \langle \Phi(x) - \gamma \mu(q_t), u_{p,q_t} \rangle q_t(x) dx$$

$$= \quad \langle \delta_{p,q_t}, C_\gamma(q_t) u_{p,q_t} \rangle + \underbrace{\gamma \langle \delta_{p,q}, \mu(q_t) \rangle \langle \mu(q_t) - \gamma \mu(q_t), u_{p,q_t} \rangle}_{=0, \text{ for } \gamma \in \{0,1\}}$$

$$= \quad \langle \delta_{p,q_t}, C_\gamma(q_t) u_{p,q_t} \rangle + 0.$$

Consider $\Psi(q) = \frac{1}{2} \text{MMD}^2(p,q) = \frac{1}{2} \|\mu(p) - \mu(q)\|^2$, it is easy to see that the functional derivative wrt to $q$ is $D_q \Psi(q)(x) = -\delta_{p,q}$. Hence we have:

$$\frac{1}{2} \frac{d\text{MMD}^2(p,q_t)}{dt} = - \int \langle \nabla_x u_{p,q_t}(x), \nabla_x \delta_{p,q_t} \rangle q_t(x) dx - \alpha \int \delta_{p,q_t}(x)(u_{p,q_t}(x) - \gamma \mathbb{E}_{q_t} u_{p,q_t}) q_t(x) dx$$

$$= - \langle \delta_{p,q_t}, D(q_t) u_{p,q_t} \rangle - \alpha \langle \delta_{p,q_t}, C_\gamma(q_t) u_{p,q_t} \rangle$$

$$= - \langle \delta_{p,q_t}, (D(q_t) + \alpha C_\gamma(q_t) + \lambda I - \lambda I) u_{p,q_t} \rangle$$

$$= -(\langle \delta_{p,q_t}, (D(q_t) + \alpha C_\gamma(q_t) + \lambda I) u_{p,q_t} \rangle - \lambda \langle \delta_{p,q_t}, u_{p,q_t} \rangle)$$

$$= -(\langle \delta_{p,q_t}, \delta_{p,q_t} \rangle - \lambda \langle \delta_{p,q_t}, u_{p,q_t} \rangle) \text{ where we used that } (D(q_t) + \alpha C_\gamma(q_t) + \lambda I) u_{p,q_t} = \delta_{p,q_t}$$

$$= -\left( \text{MMD}^2(p,q_t) - \lambda \text{SF}^2_{\mathcal{H},\gamma,\lambda}(p,q_t) \right) \text{ by Definition of Sobolev-Fisher Distance}$$

$$\leq 0$$

since

$$\text{MMD}^2(p,q_t) \geq \lambda \text{SF}^2_{\mathcal{H},\gamma,\lambda}(p,q_t)$$

$\square$

We now prove a Lemma the can be used to show that Unbalanced Sobolev descent has an acceleration advantage over Sobolev descent [2].

**Lemma 2.** *In the regularized case $\lambda > 0$ with $\alpha > 0$, the Kernel Sobolev-Fisher Discrepancy $SF_{\mathcal{H},\gamma,\lambda}$ is strictly upper bounded by the Kernel Sobolev discrepancy $\mathcal{S}_{\mathcal{H},\lambda}$ [2]:*

$$SF^2_{\mathcal{H},\gamma,\lambda}(p,q) < \mathcal{S}^2_{\mathcal{H},\lambda}(p,q).$$

*Proof.* Recall that (see Proposition 2):

$$\text{SF}^2_{\mathcal{H},\gamma,\lambda}(p,q) = \left\langle (D + \alpha C_\gamma + \lambda I_m)^{-1} \delta_{p,q}, \delta_{p,q} \right\rangle,$$

and that (see [2]):

$$\mathcal{S}^2_{\mathcal{H},\lambda}(p,q) = \left\langle (D + \lambda I_m)^{-1} \delta_{p,q}, \delta_{p,q} \right\rangle.$$

We now make use of the *Woodbury identity* $(A+B)^{-1} = A^{-1} - (A + AB^{-1}A)^{-1}$ with $A = D + \lambda I_m$ and $B = \alpha C_\gamma$, which allows us to write:

$$(D + \alpha C_\gamma + \lambda I_m)^{-1} = (D + \lambda I_m)^{-1} - E, \qquad (5)$$

where $E = (A + AB^{-1}A)^{-1}$.

Notice that $A = D + \lambda I_m$ and $B = \alpha C_\gamma$ are both symmetric positive definite (SPD). Because the inverse of a SPD matrix is itself a SPD matrix, $B^{-1}$ is SPD. Because the product of SPD matrices is itself SPD, $AB^{-1}A$ is SPD. Because the inverse of the sum of SPD matrices is itself SPD, $E$ is SPD.

Equation (5) then implies:

$$(D + \alpha C_\gamma + \lambda I_m)^{-1} \prec (D + \lambda I_m)^{-1},$$

which, together with the definitions of $\mathrm{SF}^2_{\mathcal{H},\gamma,\lambda}$ and $\mathcal{S}^2_{\mathcal{H},\lambda}$, concludes the proof. $\qquad\square$

# D  Unbalanced Sobolev Descent With a Universal Kernel

While we presented the paper in a finite dimensional RKHS, to ease the presentation. We show in this Section, that our theory is general and apply to the infinite dimensional case. Of interest to us, is the case of a universal kernel. The convergence in MMD for a universal kernel implies the weak convergence in the distributional sense.

## D.1  Kernel Mean Embeddings, Covariance and Grammian of Derivatives Operators

Let $\mathcal{H}$ be a Reproducing Kernel Hilbert Space with an associated kernel $k : \mathcal{X} \times \mathcal{X} \to \mathbb{R}^+$. We make the following assumptions on $\mathcal{H}$ as in [2]:

- A1  There exists $\kappa_1 < \infty$ such that $\sup_{x \in \mathcal{X}} \|k_x\|_{\mathcal{H}} < \kappa_1$.
- A2  The kernel is $C^2(\mathcal{X} \times \mathcal{X})$ and there exists $\kappa_2 < \infty$ such that for all $a = 1 \ldots d$: $\sup_{x \in \mathcal{X}} Tr((\partial_a k)_x \otimes (\partial_a k)_x) < \kappa_2$.
- A3  $\mathcal{H}$ vanishes on the boundary (assuming $\mathcal{X} = \mathbb{R}^d$ it is enough to have for $f$ in $\mathcal{H}$ $\lim_{\|x\| \to \infty} f(x) = 0$).

The reproducing property give us that $f(x) = \langle f, k_x \rangle_{\mathcal{H}}$ moreover $(D_a f)(x) = \frac{\partial}{\partial x_a} f(x) = \langle f, (\partial_a k)_x \rangle_{\mathcal{H}}$, where $(\partial_a k)_x(t) = \left\langle \frac{\partial k(s,.)}{\partial s_a} \big|_{s=x}, k_t \right\rangle$. Note that those two quantities ($f(x)$ and $(D_a f)(x)$) are well defined and bounded thanks to assumptions A1 and A2 [28].
Similar to finite dimensional case we define the Gramian of derivatives operator of a distribution $q$ :

$$D(q) = \mathbb{E}_{x \sim \nu_q} \sum_{a=1}^{d} (\partial_a k)_x \otimes (\partial_a k)_x \quad D(\nu_q) \in \mathcal{H} \otimes \mathcal{H} \tag{6}$$

The Kernel mean embedding is defined as follows:

$$\mu(p) = \mathbb{E}_{x \sim \nu_p} k_x \in \mathcal{H}. \tag{7}$$

The covariance operator is defined as follows for $\gamma \in \{0, 1\}$:

$$C_\gamma(q) = \mathbb{E}_{x \sim q} k_x \otimes k_x - \gamma \mu(q) \otimes \mu(q) \tag{8}$$

## D.2  Regularized Kernel Sobolev Fisher Discrepancy

Let $\lambda > 0, \alpha \geq 0$, similarly the Kernel Sobolev Fisher Discrepancy has the following form:

$$\mathrm{SF}^2_{\mathcal{H},\gamma,\lambda}(p, q) = \left\| (D(q) + \alpha C_\gamma(q) + \lambda I)^{-\frac{1}{2}} (\mu(\nu_p) - \mu(\nu_q)) \right\|^2_{\mathcal{H}},$$

where $D(q), \mu(q), C_\gamma(q)$ are defined in Equations (6),(7) and (8) respectively. The Sobolev Fisher witness function is defined as follows:

$$u^{\lambda,\gamma}_{p,q} = (D(q) + \alpha C_\gamma(q) + \lambda I)^{-1}(\mu(\nu_p) - \mu(\nu_q)) \in \mathcal{H}$$

its evaluation function is

$$u^{\lambda,\gamma}_{p,q}(x) = \left\langle (D(\nu_q) + \lambda I)^{-1}(\mu(\nu_p) - \mu(\nu_q)), k_x \right\rangle_{\mathcal{H}}$$

and its derivatives for $a = 1 \ldots d$:

$$\partial_a u^{\lambda,\gamma}_{p,q}(x) = \left\langle (D(\nu_q) + \lambda I)^{-1}(\mu(\nu_p) - \mu(\nu_q)), \partial_a k_x \right\rangle_{\mathcal{H}}.$$

### D.3 USD with Infinite dimensional Kernel decreases the MMD distance

Theorem 3 holds for the infinite dimensional case. To see that it is enough to replace in the proof of Theroem 3 finite dimensional operators and embeddings $D(q), C_\gamma(q), \mu(q)$ with their infinite dimensional counterparts given in Equation in Equations (6),(7) and (8). All norms and dot products in $\mathbb{R}^m$, are also to be replaced with $\|.\|_{\mathcal{H}}$ and $\langle \cdot, \cdot \rangle_{\mathcal{H}}$.

## E   Code and Hyper-parameters

Listing 1: Pytorch code for computing cost function $\mathcal{L}_S(\xi, \lambda)$ in Algorithm 3

```python
import torch
from torch.autograd import grad

def descent_cost(f, x_p, w_p, x_q, w_q, lambda_aug, alpha, rho, gamma=1):
    """Computes the objective of Unbalance Sobolev Descent and returns the loss = -obj
    """
    x_q.requires_grad_(True)

    f_p, f_q = f(x_p), f(x_q)
    Ep_f = (w_p * f_p).mean()
    Eq_f = (w_q * f_q).mean()

    # FISHER
    constraint_F = (w_q * f_q**2).mean() - gamma * Eq_f**2

    # SOBOLEV
    grad_f_q = grad(outputs=Eq_f, inputs=x_q, create_graph=True)[0]
    normgrad_f2_q = (grad_f_q**2).sum(dim=1, keepdim=True)
    constraint_S = (w_q * normgrad_f2_q).mean()

    # Combining FISHER and SOBOLEV constraints
    constraint_tot = (constraint_S + alpha * constraint_F - 1.0)

    obj_f = Ep_f - Eq_f \
            - lambda_aug * constraint_tot - rho/2 * constraint_tot**2

    return -obj_f, Ep_f, Eq_f, normgrad_f2_q
```

## F   Architecture of Neural Network discriminator

```
D_mlp = Sequential(
    (L0): Linear(in_features=n_inputs, out_features=n_layers[0], bias=True)
    (N0): ReLU(inplace=True)
    (L1): Linear(in_features=n_layers[0], out_features=n_layers[1], bias=True)
    (N1): ReLU(inplace=True)
    (D1): Dropout(p=0.2, inplace=False)
    (L2): Linear(in_features=n_layers[1], out_features=n_layers[2], bias=True)
    (N2): ReLU(inplace=True)
    (V): Linear(in_features=n_layers[2], out_features=1, bias=False)
)
```

## G   Hyperparameters for experiments

Listing 2: Hyperparameters for synthetic experiments (Figs. 1, 2, 5, 6)

```
{
    "n_layers": [64, 1024, 64],   # Number of neurons in hidden layers of discriminator
    "n_points_src": 4000,         # Number of points sampled from source distribution
    "n_points_target": 4000,      # Number of points sampled from target distribution
    "T": 800,                     # Number descent steps
    "optimizer": Adam(amsgrad=True) # Optimizer for discriminator (reset at every update of distribution q)
    "batchSize": 512,             # Batch size for discriminator updates
    "n_c_startup": 200,           # Number of steps for discriminator updates at startup
    "n_c": 20,                    # Number of steps for discriminator updates in-between updates of distribution q
    "wdecay": 1e-5,               # Weight decay factor
    "lrD": 1e-4,                  # Learning rate for discriminator updates
    "lrQ": 1e-4,                  # Learning rate for updates of distribution q
    "tau": 1e-3,                  # Birth-death rate
    "alpha": 0.6,                 # Damping factor ($\alpha$ in Algorithm 3)
    "lambda_aug_init": 1e-5,      # Initialization of augmented Lagrange multiplier (in Algorithm 3)
    "rho": 1e-6                   # Learning rate of augmented Lagrange multiplier
}
```

Listing 3: Hyperparameters for color transfer experiments (Figs. 3, 7)

```
{
    "n_layers": [128, 2048, 128], # Number of neurons in hidden layers of discriminator
    "n_points_src": 65536,        # Number of points sampled from source distribution
    "n_points_target": 65536,     # Number of points sampled from target distribution
    "T": 800,                     # Number descent steps
    "optimizer": Adam(amsgrad=True) # Optimizer for discriminator (reset at every update of distribution q)
    "batchSize": 500,             # Batch size for discriminator updates
    "n_c_startup": 300,           # Number of steps for discriminator updates at startup
    "n_c": 5,                     # Number of steps for discriminator updates in-between updates of distribution q
    "wdecay": 1e-5,               # Weight decay factor
    "lrD": 1e-4,                  # Learning rate for discriminator updates
    "lrQ": 1e-4,                  # Learning rate for updates of distribution q
    "tau": 1e-6,                  # Birth-death rate
    "alpha": 0.3,                 # Damping factor ($\alpha$ in Algorithm 3)
    "lambda_aug_init": 0.0,       # Initialization of augmented Lagrange multiplier (in Algorithm 3)
    "rho": 1e-6                   # Learning rate of augmented Lagrange multiplier
}
```

Listing 4: Hyperparameters for single-cell analysis interpolation experiments (Fig. 4, 8)

```
{
    "n_layers": [128, 1024, 64], # Number of neurons in hidden layers of discriminator
    "n_points_src": 3500,        # Number of points sampled from source distribution
    "n_points_target": 3500,     # Number of points sampled from target distribution
    "T": 400,                    # Number descent steps
    "optimizer": Adam(amsgrad=True) # Optimizer for discriminator (reset at every update of distribution q)
    "batchSize": 100,            # Batch size for discriminator updates
    "n_c_startup": 300,          # Number of steps for discriminator updates at startup
    "n_c": 5,                    # Number of steps for discriminator updates in-between updates of distribution q
    "wdecay": 1e-5,              # Weight decay factor
    "lrD": 1e-4,                 # Learning rate for discriminator updates
    "lrQ": 1e-4,                 # Learning rate for updates of distribution q
    "tau": 2e-4,                 # Birth-death rate
    "alpha": 0.2,                # Damping factor ($\alpha$ in Algorithm 3)
    "lambda_aug_init": 1e-5,     # Initialization of augmented Lagrange multiplier (in Algorithm 3)
    "rho": 1e-6                  # Learning rate of augmented Lagrange multiplier
    "normalization": nn.BatchNorm1d(track_running_stats=False, momentum=0.0) # Substitutes dropout layer after second hidden
        layer
}
```

# H    Additional Plots

## H.1    Synthetic Examples

We give in Figs 5 and 6 additional synthetic experiments:

## H.2    Image Coloring

We give in Figure 7, the trajectories of the descent in image color transfer experiment.

## H.3    Comparisons to Waddington Optimal Transport for single-cell analysis

We give in Figure 8 the evolution of the MMD as function of the day of interpolation using USD and unbalanced OT as in WOT.

(a) Neural Unbalanced Sobolev Descent paths in transporting a Gaussian to circles). We compare Sobolev descent (SD, [2]) to both USD implementations with birth and death processes (bd: Algorithm 2) as well as the weighted version implementation (w: Algoritm 1, note that in this case we overlay the points with their respective weights where coloring density encodes the weights). We see that birth and death processes helps USD to outperform SD in capturing the two modes.

(b) MMD function of the time in the descent from a Gaussian to Circles: We see that birth and death processes in both implementations of USD accelerate the convergence to the target distribution and reaches lower MMD than Sobolev Descent that relies on advection only.

Figure 5: Neural Unbalanced Sobolev Descent transporting a Gaussian to circles (target samples have uniform weights, $a_j = \frac{1}{n}$).

(a) Neural Unbalanced Sobolev Descent paths in transporting a disk to a heart/spiral. We compare Sobolev descent (SD, [2]) to both USD implementations with birth and death processes (bd: Algorithm 2) as well as the weighted version implementation (w: Algoritm 1, note that in this case we overlay the points with their respective weights where coloring density encodes the weights). We see that birth and death processes helps USD to outperform SD in capturing the two modes.

(b) MMD function of the time in the descent from a disk to a heart/spiral: We see that birth and death processes in both implementations of USD accelerate the convergence to the target distribution and reaches lower MMD than Sobolev Descent that relies on advection only.

Figure 6: Neural Unbalanced Sobolev Descent transporting a 'disk' to a 'heart' weighted by a spiral-shaped gradient.

Figure 7: Color Transfer with USD using (bd) Algorithm 2. Trajectories of the descent.

Figure 8: MMD and EMD between predicted mid points (using USD and WOT) and their respective ground truths as function of the day of interpolation.