[Reviews · NeurIPS 2020]

Review 1

Summary and Contributions: This paper is concerned with the design of gradient flows of particles that can change mass and position, as opposed to a standard particle flow, where only position is changed. To this goal, the authors merge two known approaches: Sobolev descent and unbalanced Optimal Transport. The contributions of this paper are mostly theoretical: in particular, they define a (rather new) integral probability metric, the Sobolev-Fisher discrepancy via the linearisation of the so-called Wassersetein-Fisher-Rao (WFR) distances. Their descent scheme is a kernel regularized approximation of the gradient flow of WFR, via Sobolev-Fisher discrepancy. They show faster convergence of their unbalanced scheme w.r.t. direct balanced Sobolev descent and it is explained by the change of mass. The authors propose a practical implementation using neural networks and they show its efficiency on synthetic datasets, mixture of gaussians, images densities and color transfer.

Strengths: This work is mostly a theoretical contribution on unbalanced flows of distributions. The use of unbalanced descent algorithms appear in different applications, such as neural networks optimization, density matching which are of recent interest for the NeurIPS community. So far I checked the results are correct. Although not unexpected, this work shows a gain in performance of the sobolev-fisher discrepancy gradient flow on standard balanced algorithm. It also gives a sound theoretical proof of it. In particular, the experiments show a clear gain in terms of speed of convergence of the unbalanced Sobolev descent over the balanced setting.

Weaknesses: - Novelty: This work can appear as incrementally innovative since Sobolev descent and unbalanced optimal transport were already present in the literature. This work is essentially the merging of these two existing works, resulting however in a new contribution. - Empirical evaluation: The experiments are still a bit toyish and even on these toyish examples do not really outperform existing results.

Correctness: The theoretical claims are correct so far I checked. On the methodological side, I do appreciate that the authors provide a comparison using both MMD and OT-like distances in their experiments for the developmental trajectories of cells, which seems a fair comparison. In the same direction, the experiment on color transfer uses an MMD distance and they show that USD achieves smaller MMD score. In my opinion, the main difference between unbalanced optimal transport gradient flow and USD gradient flows is due to the MMD regularization. If it is the same MMD that is used for evaluation, then the better results for USD are not so surprising since the evaluation metric is part of the model. I would like the authors to comment on that point in the main text.

Clarity: The paper is well-written and right to the point. It is not required to read the mentioned references for a correct understanding of the paper, although it helps to put the contribution in perspective.

Relation to Prior Work: Obviously, the difference with Sobolev descent is clearly discussed since the reaction term is introduced. The difference with unbalanced gradient flows is unclearly stated in my opinion: the authors argue « we rather learn dynamically the flow that corresponds to the witness function of the Sobolev-Fisher discrepancy ». Maybe the authors could elaborate a bit on what they mean. In my understanding of their work, the authors are simply doing a gradient descent with respect to the kernelized Wasserstein-Fisher-Rao metric since the Sobolev-Fisher is a linearisation of it and gradient flow only depends on the linearization of the discrepancy. If I misunderstood the work, then I would advise the authors to explain in more details the difference. The proposed gradient flow algorithm shares some similarities with splitting scheme in the mathematical litterature, in particular works developed for unbalanced optimal transport.

Reproducibility: Yes

Additional Feedback: For improvement (only my personal taste), I would like to see the impact of the choice of the regularising MMD, which as a limiting case includes the unbalanced optimal transport gradient flows. After reading the answers, I am not completely satisfied with their answers: on R1 Q2: in my opinion, their method is a faster computational method (up to the kernel addition) of unbalanced gradient flows. I also reckon that the experimental part is not detailed enough on the computational times, etc... However, I do not change my overall score.


Review 2

Summary and Contributions: The paper introduces Unbalanced Sobolev Descent, which applies the Sobolev-Fisher discrepancy between distributions. The authos show that this approch is related to the Wasserstein-Fisher-Rao metric between distributions. To estimate witness function the authors used neural networks and proposed two algorithms.

Strengths: The idea of the paper is interesting and the methodology is justified theorically. The authors introduce the Kernel Sobolev-Fisher (SF) discrepancy which is related to the Wasserstein Fisher-Rao (WFR). Since WFR requires to calculate PDEs (calulate integral), the authors propose another solution, which is still computationally challenging. The authors gave a variant to efficiently estimate SF using neural networks.

Weaknesses: I have no major objections to this work. The authors should explain what is the processing time of the proposed algorithms in relation to the compared methods?

Correctness: The theory and experiments are rather correct.

Clarity: Yes

Relation to Prior Work: Yes

Reproducibility: Yes

Additional Feedback: Post rebuttal ============================================== Following the author response and in seeing the concerns raised by the other reviewers. I would like to maintain my current recommendation (accept). I agree that the numerical experiments are not entirely convincing. But I think that the paper is clear and the idea is sound.


Review 3

Summary and Contributions: The authors use RKHS space to approximate the dual space of the linearized Wassertein-Fisher-Rao metric between unbalanced distributions. This approximation, named Sobolev-Fisher discrepancy, has a closed-form solution in term of samples. They apply this discrepancy function as the objective function for generative model sampling problems with applications in molecular biology.

Strengths: The method applies RKHS space to approximate locally the Wasserstein or Fisher-Rao type metric. And locally, this metric exhibits closed form solutions, which can be used as an objective function for learning.

Weaknesses: 1. This paper's idea is very closely related to Sobolev descent paper. The only difference is that the metric is changed from Wasserstein to Wasserstein-Fisher-Rao metric. 2. Can the author provide some qualitative analysis on the perspective of kernel approximations? Is there any practical guidance of the choice of kernels in RKHS? 3. The title is a little bit confusing. Usually, the Sobolev descent refers to the gradient descent of energy in this Sobolevs space, like in gradietn flow studies. In this paper, the descent is the gradient operator of the Sobolev type metric function. It may be better to emphasize the Sobolev-Fisher discrepancy.

Correctness: The claim and method are correct.

Clarity: The paper is well written. In line 140, ``We now prove a Lemma'' may change to "We now prove a lemma"

Relation to Prior Work: There are related works in approximating the gradient flows in Wasserstein or Fisher-Rao space. Arbel, et.al. Kernelized Wasserstein natural gradient. ICLR 2020. Alex, et.al. APAC-Net: Alternating the Population and Agent Control via Two Neural Networks to Solve High-Dimensional Stochastic Mean Field Games. By the way, for unbalanced optimal transport, the unnormalized OT is also related. Gangbo, et.al. Unnormalized optimal transport. arXiv:1902.03367.

Reproducibility: Yes

Additional Feedback: I have read the response. The authors answer all my questions clearly. A good paper.


Review 4

Summary and Contributions: In this work, the authors extent the so called Sobolev descent framework of [2] to the case where the source and target distributions do not have the same total mass (unbalanced case [6,7,8,9]).

Strengths: The theoretical contributions of this work are valuable though incremental. It seems well mathematically grounded. The methods used in the proof are technical, though not new.

Weaknesses: Numerical experiments are really not convincing: the authors limit themselves to a collection of poorly treated examples. Despite announcing "transporting high dimensional distribution" (line 2), there is not a single figure about computational time, size of the data (except a puzzling remark line 212: "images are subsampled for computational feasibility"). The single-cell dataset is treated without any serious scientific considerations. There is no quantitative results to assess the performance of the method. There is mainly visual checks. A very partial piece of code is written in the text, and no release is announced in the text. This makes me feel that the method was developed to write theorems but not to be used in practice.

Correctness: The theoretical developments seem to be correct, though i didn't check the proof line by line.

Clarity: The paper is dense. But, overall, it is decently written considering the 8 pages limit. Nevertheless, the text is technical (the author speaks jargon, e.g. "witness function") and focus almost only on theory. Finally, the core paper is highly not self contained and the reader need to dig into the supplementary material to understand or cited references the method (especially the algorithm).

Relation to Prior Work: nothing to report

Reproducibility: No

Additional Feedback: I think that this contribution miss his target: this is a theoretical development (using technical but standard methods of proofs) but without a real substantial practical numerical/methodological input. It will be more suited in a journal paper with some extra work on the algorithm and numerical part. ----------------------- Rebuttal -------------------------- I have read the answer. The authors will release their code which is a positive point. Nevertheless, I doubt that the promised changes will make the numerical part convincing...

[Author Response · NeurIPS 2020]

We thank the reviewers for their comments and valuable feedback. In the following we address their questions:
2

**Reviewer1**: We thank the Reviewer for several useful suggestions that we will gladly include in the revisions.
**Q1: If it is the same MMD that is used for evaluation, then the better results for USD are not so surprising**
**since the evaluation metric is part of the model.**
**A:** To make comparisons fair, the MMD used for evaluation is different from the one used at training. For evaluation,
we used a fixed MMD defined in lines 196-198: "In all our experiments we report the MMD distance with a Gaussian
kernel, computed using random Fourier features (RF) with 300 RF and a kernel bandwith equal to $\sqrt{d}$".
**Q2: The difference with unbalanced gradient flows is unclearly stated in my opinion.**
**A:** Thank you for pointing out the lack of clarity about this. We will clarify this point in the revisions. The main
difference is that unbalanced gradient flows of WFR give a full path from source to target, that requires to solve a
sequential planning problem where all distributions on the path have to be simultaneously computed. Unbalanced
Sobolev Descent on the other hand finds a path from source to target computing each step in a greedy way.
**Q3: The proposed gradient flow algorithm shares some similarities with splitting scheme.**
**A:** Thank you for pointing this out, this is indeed the case. We will add a discussion about this.
**Q4: I would like to see the impact of the choice of the regularizing MMD.**
**A**: Thank you for the suggestion. We will be glad to add this ablation experiment for the synthetic examples.
18

**Reviewer2**: Thank you for suggestions that we will also address in the revised paper.
**Q: Explain what is the processing time of the proposed algorithms in relation to the compared methods.**
**A:** Thank you for pointing out an opportunity to clarify our work. We gave computational complexity of Neural USD in
lines 178-182, and said that it scales linearly in samples: $O(N)$. But we will be glad to give more explicit comparisons
to the OT Baselines in the revisions. Entropic regularized unbalanced OT solved with Sinkhorn that were used in the
paper are expensive computationally as they scale as $O(N^2)$, where $N$ is number of samples. Better scaling alternatives
in OT with sample size complexity that rivals our method ($O(N)$) exist, but either require approximation or more
complex optimization schemes. Examples are Altschuler et al. that relies on a Nystrom approximation, and Genevay et
al. that relies on stochastic optimization. We will add this discussion to the paper.

**Reviewer3**: Thank you for your feedback and useful literature pointers that we will gladly add to the revised paper.
**Q1: This paper's idea is very closely related to Sobolev descent paper. The only difference is that the metric is**
**changed from Wasserstein to Wasserstein-Fisher-Rao metric.**
**A:** That is indeed the main formal difference compared to Sobolev descent. This slight formal change however results
in several qualitative differences such as the fact that corresponding transport now include a reaction term, in addition to
advection. The main practical implications are that we can therefore diffuse to and from distributions of different mass,
we can deal with weighted samples, and that the descent is provably accelerated, which we also show empirically.
**Q2: Is there any practical guidance of the choice of kernels in RKHS?**
**A:** For the kernel one can use the consolidated heuristic to select the bandwidth of the Gaussians based on the median
distance between samples in the source and target distributions.
**Q3: The title is a bit confusing. A:** Thanks for pointing this out. We'll try to highlight the Sobolev-Fisher Discrepancy.
**Q4: Additional references. A:** Thanks for all the suggested citations, we will make sure to add them to the main text.

**Reviewer4**: We thank the Reviewer for their time. We believe we can address most misunderstandings by R4.
**Q1: There is no quantitative results to assess the performance of the method. A:** We respectfully disagree. Figs
1b, 2b, give quantitative results, i.e. the MMD of the resulting distribution found with USD to the target. In Fig 3, the
MMD to the target is also given under each image. Fig 4 quantitatively evaluates OT vs USD on a cell biology dataset.
**Q2: The single-cell dataset is treated without any serious scientific considerations.**
**A:** This is a theoretical and methodological paper. We don't think it is our place to draw scientific conclusions about
biology experiments. This is the role of domain experts. Our goal was to demonstrate the potential of our method to
analyse this type of data, and for that we carefully followed the same experimental protocol of Schiebinger et al.
**Q3: There is not a single figure about computational time, size of the data.**
**A:** We assume R4 might have overlooked section "Computational and Sample Complexities" (line 178) where we
discuss this. We'll gladly add explicit comparisons with OT that tends to scale worse than our method (in fact, the
comment on line 212 singled out by R4 was about OT, known to require subsampling for feasibility, unlike our method).
**Q4: A very partial piece of code is written in the text. A:** We'll release code and analyses upon acceptance.
**Q5: The author speaks jargon, e.g. "witness function". A:** We respectfully point out that "witness function" is an
accepted term, commonly adopted in IPM papers and now permeating even the GANs literature.
**Q6: this is a theoretical development but without a real substantial practical numerical/methodological input.**
**A:** We disagree. We introduce a novel kernelized discrepancy and show how its flow can be used in practical settings
for unbalanced transport. Our algorithm is numerically faster than conventional OT methods and previous alternatives.

[Meta-Review · NeurIPS 2020]

The overall opinion is that this paper provides a valuable theoretical contribution, and I recommend accept. However, some reviewers (R4) have expressed concerns about the practical and implementation aspect of the work, and it is important that the authors implement the updates that they have promised in the rebuttal for the final version. Moreover, the lack of clarity raised by reviewer 1 (Q2) was not satisfyingly addressed (see his/her updated review: you are actually computing the same gradient flow (up to the kernelized distance), but the advantage of your method is that it is computationally cheaper than WFR). It is important to take those remarks into account in the final version to write a rigorous discussion.